# Impact of electronic health records on nursing workflow efficiency and predictive factors in Palestinian hospitals

Fuad Farajalla ⓘ*, Mousa Farajallah, Nesreen Alqaissi, Mohammad Qtait ⓘ,
Zeenat Mousa Mesk

Palestine Polytechnic University, Hebron, Palestine

* Fuad.farajalla@gmail.com

## Abstract

Electronic Health Records (EHRs) have revolutionized patient care and data management, but their integration may disrupt workflow. Palestine recently introduced EHRs in their hospitals, yet no local data exist on nursing workflow. This study aims to assess the impact of EHRs on workflow efficiency and associated factors among nurses with direct paper-to-EHR transition experience. A quantitative, cross-sectional study design was employed. A convenience sample of 185 nurses was recruited from medical and surgical wards across selected hospitals. Data were collected via a structured questionnaire and analyzed using SPSS version 29 using descriptive statistics and multiple linear regressions, with significance set at p<0.05. A total of 185 nurses participated, with the majority aged 25–34 years (61.6%). Most had 5–10 years of experience (42.2%). Overall, 70% of nurses reported high or very high workflow efficiency with EHR use (M=3.59, SD=0.75). EHRs were perceived to improve access to patient information (63.2%), reduce documentation time (63.8%), and support teamwork and communication, although 50.8% reported workflow interruptions due to technical issues. Multiple regression identified EHR user-friendliness (β=0.261, p<0.001), training (β=0.243, p=0.024), technical support (β=0.184, p=0.005), and age (β=–0.223, p=0.037) as significant predictors of workflow efficiency. EHRs positively influence nurses' workflow; transition-experienced nurses highlight usability and training as significant predictor factors. Enhancing these areas can optimize clinical performance.

## Author summary

This is the first study conducted in Palestine to explore the impact of Electronic Health Records (EHRs) on nurses' day-to-day work in government hospitals. This study was conducted in the context of political tension, regular power outages, and limited resources. 185 nurses who transitioned directly from a paper-based record system to EHRs responded to the survey.

**Data availability statement:** The data supporting the findings of this study are presented in the manuscript tables and are freely available in full within the supplementary files for use by other researchers.

**Funding:** The author(s) received no specific funding for this work.

**Competing interests:** The authors have declared that no competing interests exist.

The vast majority (70%) of nurses indicated EHRs enhance their efficiency of work through faster access to patient information and reduced time spent on paperwork. Many respondents (50%) also experience organizational delays due to the system's slow progress. Key organizational factors enhancing EHR efficiency include good usability, training, technical support, and younger age.

The findings demonstrate that digital technologies and digital systems can provide tremendous efficiencies for nurses even in extreme situations, but that successful implementation is contingent upon finding solutions to basic infrastructural challenges. Hospitals can use findings from this study to better target nurse-friendly implementations of EHR systems, ongoing education and training, and reliable backup systems, as these changes could help decrease stress and burnout as well as improve patient care. Our research contributes practical background information to an ongoing global discussion on EHRs in low-resource settings while also emphasizing the continued challenge of ensuring health technologies support frontline staff and care.

## Introduction

The provision of health services to a population is a complex process that depends greatly on health information. Electronic Health Records (EHRs) are computerized patients' medical records that enable easier storage and access to patient information using computers [1]. EHRs generally integrate the critical data of patients along with their corresponding treatment outlines. These systems were developed in the 20th century and have since evolved into sophisticated clinical management systems [2]. Workflow is the structure of tasks grouped together in a sequence of events producing a process, along with the people involved and resources to achieve a desired outcome [3]. EHRs can improve patient outcomes and the quality of care, decrease medication errors, minimize unnecessary tests, and enhance care coordination by improving the ability to access information [4,5].

However, despite these advantages, EHRs are associated with limitations that require attention [1]. Nursing workflows are often hindered by complications, such as issues with technology, constant distractions, and a lack of access to basic tools [6]. If interruptions exceed the time allocated for specific activities, they can hinder progress, waste time, and compromise the quality of care [6,7]. These issues can result in increased nurse stress, burnout, dissatisfaction, and higher mental workload [8].

Nurses devote more than half of their shift time to EHR system data entry and retrieval [9]. With the adoption of EHR systems in healthcare organizations, there is an increasing need to create evaluation frameworks that enable comparison of their effectiveness from the users' perspective. However, evaluation of EHR systems has received little attention from nurses who deliver round-the-clock patient care.

Worldwide, the impact of EHRs on the quality of healthcare is well understood [10,11]. Advanced countries, including the US, effectively integrated EHRs into their respective healthcare setups, depicting benefits in terms of the accessibility of information and efficiency in workflow processes. However, complexity is still one problem with the system [12]. What's more, developing countries, including Bangladesh, Indonesia, Libya, and Ethiopia, have also adopted EHRs, with studies depicting the benefits as well as the challenges [13,14]. While in the Indonesian context, the utilization of EHRs has decreased the nurses' workloads in the administrative department [14].

In the Middle East, regional studies had indicated the disparity in the usage and effectiveness of EHR. For instance, many hospital settings in Jordan still rely on the paper format with respect to the collection of patients' data, hence wasting time that could otherwise be spent on attending to the patients [15]. In Saudi Arabia, even with the intense investment in the healthcare system, there is the issue of efficiency with respect to the system's usability [2].

In Palestine, there is prolonged political and economic instability in healthcare [16,17]. Recent Palestinian evidence shows that system-level barriers such as equipment failure, unfamiliarity with emergency trolleys, team stress, and lack of leadership significantly impair the quality of life-saving interventions [18,19]. The healthcare system also lacks sufficient nursing staff, and the nurses face challenges regarding continuing education and lack of access to electricity or the internet, among others, affecting the quality of care given to the patients [20,21]. Moreover, the nurses are facing challenges in the availability of technology support to enable the effective use of electronic healthcare records [22]. The absence of well-integrated documentation systems creates inefficiencies in nurses' workflows and impacts quality of care.

In Palestine, EHRs were introduced in hospitals only in recent years, with governmental hospitals leading adoption. Private and many rural hospitals continue to rely primarily on paper-based or hybrid systems [23]. No prior study has evaluated their impact on nursing workflow. This study is the first in Palestine to examine EHR impact on nursing workflow efficiency and predictive factors in Palestinian governmental hospitals, focusing exclusively on nurses who transitioned from paper-based to electronic systems. The study fills a critical evidence gap in a resource-constrained, recently digitized setting.

## Methods

### Ethics statement

Ethical approval was obtained prior to conducting the study from the Ethical Committee of the Nursing College, Palestine Polytechnic University (Approval No. EA/2025/48), and the Palestine Ministry of Health. All data were de-identified immediately following collection and stored securely in password-protected electronic databases in compliance with international data protection standards equivalent to GDPR requirements. Data access was restricted to the research team members only. Before data collection, written informed consent was received from all participants, and all nurses received a detailed explanation of the study's purpose, procedures, potential benefits, and risks, and each provided written informed consent. Participation was entirely voluntary, with the option to withdraw at any point without any negative consequences. Other issues included ensuring privacy during data collection, which we ensured by obtaining informed consent and conducting the data collection in a comfortable and private space. Participants received assurances that their personal information would be kept secure and confidential. The study adhered to ethical principles outlined in the Declaration of Helsinki regarding the rights, safety, and well-being of research participants.

### Design and setting

A quantitative cross-sectional design was used to investigate the impact of EHR systems on workflow efficiency and predictive factors. This study is reported in accordance with the Strengthening the Reporting of Observational Studies in Epidemiology (STROBE) guidelines for cross-sectional studies.

The study was conducted in the West Bank of Palestine, encompassing the northern, central, and southern areas. Convenience criteria were used to identify the six governmental hospitals sampled, which allowed for representation from

each region. Two hospitals from the northern region, two from the central region, and two from the southern region were identified. The selected hospitals had incidents of relevant units that aligned with the study's operational objectives and had fully implemented EHR systems.

All selected hospitals operate the national EHR system known as *Avicena*, which is the standardized platform adopted across all governmental hospitals in Palestine. Avicena is a locally developed system customized to the clinical and administrative workflows of the Ministry of Health. The system was rolled out progressively between 2018 and 2020, and all included hospitals have been fully using Avicena for at least three years prior to data collection. Core modules available to nurses include patient registration, nursing assessments and progress notes, medication administration records, vital-sign charting, laboratory and radiology ordering with result retrieval, and discharge documentation. Nursing workflows in these hospitals are predominantly fully digital, with paper documentation used only during system outages or for temporary bedside notes in a minority of units [23,24].

## Population and sampling

The participants of the study were nurses in the medical and surgical departments in governmental hospitals in Palestine. The accessible population for this study included nurses employed in medical or surgical wards in select governmental hospitals in Palestine.

Data on the number of nurses were obtained upon direct contact with the nursing directors at the governmental hospitals, as there were no formal reports available. And there are approximately 350 nurses currently working in medical and surgical departments in the northern, central, and southern areas of the West Bank. The sample size was determined using G*Power 3.1 for multiple linear regression with seven predictors (see Table 1), assuming a medium effect size ($f^2 = 0.15$), $\alpha = 0.05$, and 95% power, resulting in a minimum required sample of 150 participants. To further validate the sample size, the Raosoft sample size calculator was used, yielding a minimum of 184 based on a population of 350, a 95% confidence level, a 5% margin of error, and a 50% response distribution. To account for potential dropouts and statistical errors, the sample size was increased to 210. Non-probability convenience sampling was used to recruit participants.

## Inclusion and exclusion criteria

Registered nurses and licensed practical nurses with at least 3 months of experience using EHR systems to ensure familiarity with the system were included in the study. Additionally, participants must have previously worked with a paper-based documentation system for a minimum of three months before transitioning to the EHR system; this allows for reflection on changes in workflow and documentation quality.

## Exclusion criteria

Nurses in units that still use a mix of paper-based and electronic documentation were excluded to avoid confusion. Additionally, non-clinical staff (administrative & managerial staff) was excluded to ensure the responses reflect hands-on nursing experience.

## Instrumentation

The survey was designed for the purposes of the study based on an intensive literature review to ensure relevance to the study objectives. The survey had two main parts. The demographic part covered variables such as age, sex, number of working years, EHR training, user-friendliness, IT support, and access to computers, among others. The variables had been covered in an intensive review of literature from other studies, which had shown their importance in EHR usage in healthcare facilities. For example, user-friendliness was emphasized in a study conducted in the U.S. [25]. Similarly, studies from Saudi Arabia have shown the importance of training and technical support in improving nurses' ability to effectively use EHR systems [2].

**Table 1. Distribution of Demographics and work environmental factors information among participant.**

| Demographics | Categories | n | % |
|---|---|---|---|
| **Age/ Years** | < 25 | 10 | 5.4 |
| | 25-34 | 114 | 61.6 |
| | >35 | 61 | 33 |
| **Gender** | Male | 100 | 54.1 |
| | Female | 85 | 45.9 |
| **Experience/ Years** | <5 | 48 | 25.9 |
| | 5-10 | 78 | 42.2 |
| | >10 | 59 | 31.9 |
| **Training on EHR** | Yes | 102 | 55.1 |
| | No | 83 | 44.9 |
| **IT support** | Strongly disagree | 3 | 1.6 |
| | Disagree | 39 | 21.1 |
| | Neutral | 55 | 29.8 |
| | Agree | 67 | 36.2 |
| | Strongly agree | 21 | 11.3 |
| **Perceived usability of EHR (User-friendliness)** | Strongly disagree | 10 | 5.4 |
| | Disagree | 19 | 10.3 |
| | Neutral | 57 | 30.8 |
| | Agree | 70 | 37.8 |
| | Strongly agree | 29 | 15.7 |
| **Access to computers** | Strongly disagree | 6 | 3.2 |
| | Disagree | 29 | 15.7 |
| | Neutral | 70 | 37.8 |
| | Agree | 67 | 36.2 |
| | Strongly agree | 13 | 7.1 |

Note. n = 185.

The second section assessed nurses' perceptions of workflow efficiency when using the EHR system. This section consisted of 9 items distributed across four domains: (1) access to patient information and time management (2 items), (2) interruptions and distractions (2 items), (3) coordination of care (2 items), and (4) tracking and viewing patient progress and documentation accuracy (3 items). The exact wording of all items is provided in S1 File. All items were rated on a 5-point Likert scale ranging from 1 (strongly disagree) to 5 (strongly agree). One negatively worded item related to technical interruptions was reverse-coded prior to analysis to ensure consistent directional interpretation.

Workflow efficiency score was calculated for each participant by computing the mean of the 9 items after reverse coding. Higher scores indicated greater perceived workflow efficiency. The use of a single global mean score was justified by the high internal consistency of the scale (Cronbach's $\alpha = 0.88$) and its conceptualization as a measure of overall workflow efficiency. Although the items cover related aspects of workflow, the primary objective of this study was to assess overall efficiency rather than distinct subdomains. Therefore, a factor-structure analysis was not performed. For inferential analysis, the continuous mean workflow efficiency score (range 1–5) was retained and used as the dependent variable in the multiple linear regression model.

For descriptive interpretation only, the mean workflow efficiency scores were additionally categorized into five interpretive levels using equal interval classification of the 5-point Likert scale as recommended by Pimentel (2019) to reduce subjectivity in interpretation. Accordingly, mean scores of 1.00–1.79 were classified as very low efficiency, 1.80–2.59 as

low efficiency, 2.60–3.39 as moderate efficiency, 3.40–4.19 as high efficiency, and 4.20–5.00 as very high efficiency. This classification approach has been applied in previous studies.

The questionnaire was developed originally in English. As English is the standard language of nursing education and clinical documentation in Palestinian governmental hospitals, no translation was required.

### Validity and reliability

The validity of the survey was determined via a content validity review. To ensure a comprehensive validity review, the instrument was reviewed by two professors specializing in nursing and clinical research, along with a health informatics and computer systems expert. Modifications to the questions were made based on their expert opinions. The questions were re-evaluated for content validity after being revised.

A pilot study was conducted between **May 20 and May 30, 2025**, on 25 nurses working in medical and surgical units to assess clarity, understanding, and feasibility. Additional changes were modified based on feedback gathered from the pilot study. Changes included reorganizing tables to enhance clarity. To avoid bias or contamination, pilot study participants were excluded from the final sample. Cronbach's alpha coefficient was then used on the final sample and was reported at **0.88**, indicating high internal consistency. The complete English version of the survey is included in S1 File.

### Data collection

After ethical approval was obtained, meetings were arranged with head nurses in each ward in the selected hospitals to explain the purpose of the study and request a list of the unit participant's characteristics and their experiences, as well as obtain access to medical and surgical nurses who are eligible to participate in the study. The agreed participants were asked to sign a written informed consent form and complete a self-administered, paper-based questionnaire. The research team remained available during completion to clarify any questions and ensure proper understanding of the items. To avoid loss of the questionnaires, the research team collected the questionnaires at the end of each working day. The data was collected during the period of 15 June and 20 July of 2025, at the selected hospitals. The questionnaires were completed in the English version, and each questionnaire was coded to keep track of the number. A total of 210 questionnaires were distributed across governmental hospitals in the West Bank. Out of these, 185 were completed and returned, resulting in a response rate of 88.0%.

### Data analysis

The analysis of the data was performed with SPSS version 29. Descriptive statistics, including frequencies, percentages, means, and standard deviations, were used to summarize demographic and work environment variables. Reversed items were recoded prior to calculating total and subscale scores. A multiple linear regression analysis was performed to determine significant predictors of workflow efficiency. Age and years of experience were entered as continuous variables (years). Gender (0 = female, 1 = male) and EHR training (0 = no, 1 = yes) were binary-coded. Perceived user-friendliness, IT support, and access to computers were measured on 5-point Likert scales and treated as continuous predictors. The assumptions of normality (Shapiro-Wilk p = 0.12) and no multicollinearity (all VIF < 3.0) were met. Statistical significance was set to P < 0.05 for all analyses.

## Results

The study included a sample of 185 nurses. The largest age group of participants was 25–34, represented by 61.6% of the group. Slightly more than half of the participants were male (54.1%). All participants had prior experience using both paper and HER, which allowed for more authentic transition-based perspectives. When considering professional experience, 42.2% of participants had 5–10 years. More than half of the nurses (55.1%) reported that they had received training

on the use of EHRs. In terms of work environment factors, a good proportion of participants agreed positively with EHR usability, access to computers, and access to IT support. Supportive work environment factors were most concentrated around the "agree" and "neutral" categories, with few participants reporting negatively. Table 1 presents a summary of participant demographics and work environment factors.

A majority of respondents agreed and strongly agreed that the EHR system enhances access to patient information (63.2%) and reduces documentation time (63.8%). A similar level of agreement was observed concerning its support in tracking patient progress and its positive influence on teamwork and communication. Despite these positive perceptions, challenges remain; a total of 50.8% agree and strongly agree that they experience interruptions in workflow due to technical issues such as system crashes and slow response times (Table 2).

Among participating nurses (n = 185), the mean perceived workflow efficiency score was 3.59 (on a 5-point Likert scale), and the distribution of perceived workflow efficiency level of EHRs among the participants showed that a total of 70% of participants perceived the high or very high workflow efficiency influence of EHRs. (Table 3)

Multiple linear regression (R² = 0.35, adjusted R² = 0.32, F(7,177) = 13.45, p < 0.001) identified four significant predictors (Table 4). The regression diagnostics indicated no multicollinearity among the independent variables, with Variance Inflation Factor (VIF) values ranging from 1.12 to 2.41, well below the critical threshold. EHR user-friendliness had the strongest effect (β = 0.261, p < 0.001), followed by training (β = 0.243, p = 0.024), technical support (β = 0.184, p = 0.005), and age (β = -0.223, p = 0.037). Gender, years of experience, primary shifts, and computer availability had no significant effects (p > .05).

## Discussion

According to this study, most nurses perceive that the use of EHR improved workflow efficiency, primarily for obtaining patient information and decreasing documentation time, but also pointed out the challenges of continuing technical problems, such as system crashes.

The majority of participants were aged 25–34 years (61.6%), and there was an almost equal representation of male and female. This demographic profile is consistent with global patterns observed in healthcare settings, where younger nurses, often more adept with digital tools, dominate front-line care roles [1,10]. More than half of the respondents reported that they had received training about the EHR system (55.1%), and the system was user-friendly to 53.5%. Satisfaction decreases when nurses are asked about the availability of IT support, which is 47.5%, and access to a computer, which is 43.3%. These challenges align with what previous studies by Hariyati et al. [26] and Alsulaiman et al. [2], respectively, emphasized: that successful EHR implementation largely depends on training and ease of system use. Palestine's resource-constrained setting, characterized by unstable electricity and limited budgets, thus exacerbates these challenges by slowing down its adoption and persistent inefficiencies [27]. Such findings corroborate Naamneh and Bodas's [28] assertion that while EHRs improve efficiency, healthcare professionals still face real-life challenges in using EHRs, mainly technical issues and disruptions in workflow. Our study contradicts those emerging from developed countries, like Melnick et al. [12], which highlighted system complexity as the major concern. Therefore, infrastructure issues were considered as one of the most important barriers in Palestine that needs to be addressed through targeted interventions, such as increased technical support, a reliable power supply, and extensive internet connectivity.

One of the most distinctive findings of this study is that over half of nurses (50.8%) reported frequent workflow interruptions due to technical problems, including power outages, slow system performance, crashes, and unstable internet connectivity. Unlike studies in developed countries that emphasize system complexity and documentation burden as the main barriers (Melnick et al., 2021; Moore et al., 2020), the challenges in this Palestinian context are primarily infrastructural. Persistent electricity and internet instability undermine system reliability, disrupt continuity of care, and limit the full benefits of EHRs despite adequate usability and training. This highlights that in low-resource and conflict-affected settings, successful EHR implementation depends as much on infrastructure stability as on system design and user competence.

**Table 2. Participant responses on workflow efficiency items.**

| Workflow efficiency items | Strongly disagree | Disagree | Neutral | Agree | Strongly agree | M± SD |
|---|---|---|---|---|---|---|
| | n (%) | n (%) | n (%) | n (%) | n (%) | |
| **Access to patient information & time management** | | | | | | |
| 1. The Electronic Health Record system allows me to access and retrieve patient records **quickly and easily** than with the paper documentation. | 13 (7.0) | 19 (10.2) | 36 (19.4) | 74 (40.0) | 43 (23.2) | 3.62±1.15 |
| 2. The Electronic Health Record system has reduced the time I spend on documentation tasks compared to paper documentation. | 5 (2.7) | 28 (15.1) | 34 (18.4) | 75 (40.5) | 43 (23.3) | 3.66±1.07 |
| **Interruptions and Distractions** | | | | | | |
| 3. I experience fewer interruptions in my workflow when using the Electronic Health Record system compared to paper documentation. | 6 (3.2) | 23 (12.5) | 62 (33.5) | 64 (34.6) | 30 (16.2) | 3.39±1.00 |
| *4.Technical issues (e.g., system crashes, slow response) frequently delay my documentation tasks when using the Electronic Health Record system. | 15 (8.1) | 25 (13.5) | 52 (28.1) | 72 (38.9) | 21 (11.4) | 3.29 ± 1.11 |
| **Coordination of care** | | | | | | |
| 5. The Electronic Health Record system improves teamwork and coordination in patient care compared to paper documentation. | 8 (4.3) | 15 (8.1) | 48 (25.9) | 78 (42.2) | 36 (19.5) | 3.66±0.88 |
| 6. The Electronic Health Record system facilitates faster and clearer communication of patient updates compared to paper documentation. | 5 (2.7) | 15 (8.1) | 50 (27.0) | 76 (41.1) | 39 (21.1) | 3.71 ± 0.98 |
| **Tracking patient progress** | | | | | | |
| 7. I find it easier to track patient progress and outcomes using the Electronic Health Record system compared to paper documentation. | 5 (2.7) | 14 (7.6) | 52 (28.1) | 77 (41.6) | 37 (20.0) | 3.65±0.84 |
| 8. The Electronic Health Record system provides timely alerts about changes in patient condition. | 2 (1.1) | 21 (11.4) | 52 (28.1) | 78 (42.1) | 32 (17.3) | 3.63 ± 0.93 |
| **Accuracy of documentation** | | | | | | |
| 9. The Electronic Health Record system helps me complete documentation more accurately compared to paper documentation. | 9 (4.9) | 14 (7.6) | 50 (27.0) | 71 (38.4) | 41 (22.1) | 3.66±1.05 |
| **Total overall influence score** | (M± SD) 3.5959±.75098 | | | | | |

Note. n =185; M = mean, SD = standard deviation

*Reverse Coded

**Table 3. Perceived Level of Workflow Efficiency.**

| Perceived level of workflow efficiency categories | Mean cut of point | n | % |
|---|---|---|---|
| Very low efficiency | 1 – 1.79 | 6 | 3.2% |
| Low workflow efficiency | 1.80 – 2.59 | 16 | 8.7% |
| Moderate workflow efficiency | 2.60 – 3.39 | 33 | 17.8% |
| High workflow efficiency | 3.40 – 4.19 | 88 | 47.6% |
| Very high workflow efficiency | 4.2 – 5 | 42 | 22.7 |
| Total | | 185 | 100% |

Note. n =185.

The current study revealed that 70.3% of the nurses reported having high- or very high-level perceptions of the efficiency of workflow with the application of EHRs, with M = 3.59. Most nurses believed that the application of EHRs assisted them in obtaining easy access to the patients' records, and the time required for the documentation process was reduced. Similar benefits have also emerged from the study conducted by Moore et al. [10], along with Bauer et al. [29], wherein

**Table 4. Multiple linear regression result for predictors of workflow efficiency.**

| Predictor | B | SE | β | t | p | 95% CI for B |
|---|---|---|---|---|---|---|
| Age | −0.031 | 0.015 | −0.223 | −2.10 | 0.037 | −0.060, −0.002 |
| Gender | −0.001 | 0.045 | −0.002 | −0.02 | 0.982 | −0.089, 0.087 |
| Experience years | −0.014 | 0.012 | −0.088 | −1.15 | 0.250 | −0.038, 0.010 |
| Training on EHR | 0.186 | 0.081 | 0.243 | 2.28 | 0.024 | 0.025, 0.347 |
| Usability of EHR (User-friendliness) | 0.311 | 0.067 | 0.261 | 4.68 | <0.001 | 0.179, 0.443 |
| IT support | 0.154 | 0.054 | 0.184 | 2.86 | 0.005 | 0.049, 0.259 |
| Access to computers | 0.072 | 0.052 | 0.082 | 1.39 | 0.167 | −0.030, 0.174 |

Note. n=185; SE = standard error; β = standardized regression coefficient; t = t-value

EHRs, automated, or artificial intelligence processes increase the efficiency of obtaining information, thereby assisting the healthcare providers in managing the problem of work overload efficiently. However, about 50.3% of the nurses reported drawbacks to the workflow due to technical problems. Such problems are well stated by the current local context related to the adoption of AI technology, wherein a lack of hands-on practical experiences with technology makes the nurses concerned about the future impacts on workflow, jobs, etc. This is supported by the study conducted by Batran et al. [30], wherein lack of practical experiences with technology aggravates the concerns of nurses about workflow, jobs, etc. The adjustment with the current Palestine context indicates that due to the political instability in Palestine, the problem with lack of efficient, stable, electronic support; poor or lack of availability of the required network support; poor quality; lack of required technology support; etc., impedes the benefits of HER [16,22].

Four important variables emerged from the data that predicted workflow efficiency with EHR implementation: the user-friendliness of EHR, EHR training, EHR technical support, and age. Among these, the user-friendliness of EHR (p<0.001) and EHR training (p=0.024) are the most influential variables. The importance of having an efficient system design with efficient user training is confirmed in current studies, which emphasize the importance of system success with EHR implementation, among other variables. These findings are consistent with Alsulaiman et al. [2] and Hariyati et al. [26], who highlighted that having an EHR designed with good user interface aspects, coupled with proper training, is associated with higher user satisfaction and efficient workflow. The finding of the current study emphasizes the importance of system maintenance, especially in Palestine, where there are many challenges with system infrastructure. Age was a significant negative predictor of workflow efficiency (β=−0.223, p=0.037). This was consistent with other studies conducted by Dubale et al. [1] and Moore et al. [10], indicating that higher age was associated with lower perceived workflow efficiency when using EHR systems. Although the implication is clear, there is also the possibility of other variables, such as lack of exposure to technology or the level of work intensification. Young nurses, due to recent training or adaptation to technology, are likely to be more adaptable to EHR technology, amid other considerations presented by Pinovich et al. [25]. Using self-reported information also presents the possibility of social desirability or inaccuracy of recalling the work experiences.

Gender (p=0.982) and years of experience (p=0.250) were not significant predictors of workflow efficiency, indicating that EHR-related performance is driven primarily by system-level factors such as usability, training, and technical support rather than by demographic characteristics. The standardized nature of EHR use likely minimizes differences related to gender and professional seniority.

## Strengths and limitations

This study is the first to explore the impact of EHR on the efficiency of nursing workflows in governmental hospitals in Palestine. The study uses sound research methodology, with the sample size accurately determined by the G*Power

software and the instrument highly reliable, with an Alpha value of 0.88. The study is conducted in health facilities with fully developed EHR systems, hence providing clear, untainted information on the effect of technology on nursing work, with crucial predictive variables to enlighten the process of EHR implementation.

Despite these strengths, several limitations must be acknowledged. The cross-sectional design limits causal inference, as associations cannot be interpreted as cause-effect relationships. The use of non-probability convenience sampling introduces selection bias, as participation depended on nurses' availability and willingness to respond. This approach may also have unintentionally over-represented nurses who are more comfortable with or more positive toward technology, which could influence perceived workflow efficiency. Because the sample was drawn exclusively from governmental hospitals, the findings may not be generalizable to private hospitals, rural facilities, or institutions operating hybrid paper–electronic systems, where technological infrastructure, digital readiness, and workflow patterns may differ substantially. The use of self-reported measures raises the potential for social desirability and recall bias, and unmeasured contextual factors, such as variations in hospital resources, staffing levels, or exposure to technology, may also have influenced responses. Finally, because both predictors and outcomes were collected using the same self-reported questionnaire, common-method variance may have influenced the observed associations. In addition, workflow efficiency reflects nurses' perceived efficiency rather than objectively measured performance, as no time-motion observations or EHR log data were available.

## Implications and recommendations

The study offers practical recommendations for the optimal application of EHRs in Palestinian governmental hospitals, even in resource-limited health facilities.

## Practical and clinical approaches

- Mitigate the lack of power or access to computers by installing common workstations, backup power sources, or scheduling maintenance outside core hours.

- Improving EHR design to reduce redundant clicks and simplify navigation to save nurses' time and improve workflow.

- Offer more on-site trainings; training does not necessarily have to be technical skills only but time-saving advice and learning how to troubleshoot common issues.

## Policy and organizational measures

- Emphasize usability (strongest factor, $\beta = 0.261$) by developing national standards, with nurse-driven updates testing on the system.

- Focus on increased training sessions with the availability of technical support around the clock for system crashes and slow system response.

- Infrastructure enhancements, such as increasing computer accessibility and ensuring constant electricity and internet connectivity, are required to maximize EHR gains.

## Education

- Collaborate with local nursing colleges to integrate EHR modules into their curriculum for readiness before employment.

**Research**

- Conduct longitudinal and mixed-methods studies to assess changes in workflow efficiency and nurse satisfaction over time and mixed-method approaches to explore nurses' lived experiences with EHRs, well-being, challenges, and also cost-effective strategies for widening accessibility.

- Multi-site research across Palestinian hospitals, including private and rural facilities, can provide practical evidence to guide scalable and context-specific EHR interventions.

These measures can all work together to reduce nurse burnout, improve accuracy in documentation, and enhance patient care delivery within government hospitals.

## Conclusions

The study points out the twofold effect of EHRs on nursing practices occurring in Palestinian governmental hospitals. EHR systems improve efficiency in nursing practices by facilitating greater access to patients' data, cutting the time spent on paperwork, and facilitating coordination between healthcare providers, with the greatest number of nurses experiencing high or very high efficiency levels. Technical problems, including system failures and slower system response, experienced by 50% of nurses, are some impediments to achieving these benefits. System usability, training, technical support, and younger age were all statistically significant predicted variables. System design enhancement (ease of use) as well as improved training were targeted interventions that could optimize EHR implementation and improve nursing workflow and patient care in a resource-limited context such as Palestine.

## Supporting information

**S1 File. Data collection sheet.** The full data collection instrument used in the study, including all demographic items and workflow efficiency questions.
(DOCX)

**S2 File. STROBE checklist.** A completed STROBE (Strengthening the Reporting of Observational Studies in Epidemiology) checklist detailing adherence to reporting guidelines for cross-sectional studies.
(DOCX)

## Acknowledgments

The author thanks the nurses who participated in this study.

## Author contributions

**Conceptualization:** Nesreen Alqaissi, Mohammad Qtait, Zeenat Mousa Mesk.

**Data curation:** fuad farajalla.

**Formal analysis:** fuad farajalla.

**Funding acquisition:** fuad farajalla.

**Investigation:** Nesreen Alqaissi.

**Methodology:** Nesreen Alqaissi, Mohammad Qtait.

**Project administration:** Nesreen Alqaissi, Zeenat Mousa Mesk.

**Resources:** Mohammad Qtait.

**Validation:** Mousa Farajallah.

**Writing – original draft:** Zeenat Mousa Mesk.

**Writing – review & editing:** Mousa Farajallah, Mohammad Qtait.

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
