## [Decision Letter · Decision Letter 0]

2 Dec 2025

Response to Reviewers
Revised Manuscript with Track Changes
Manuscript
**Journal Requirements:**

1. Please ensure that your Ethics Statement is available in its entirety at the beginning of your Methods section, under a subheading 'Ethics Statement'.

2. We have noticed that you have uploaded Supporting Information files, but you have not included a list of legends. Please add a full list of legends for your Supporting Information files after the references list.

3. In the online submission form, you indicated that “The datasets used and analyzed during the current study are available from the corresponding author upon reasonable request, subject to ethical approval and institutional guidelines. To ensure long-term data stability and accessibility, requests may also be directed to the Institutional Review Board (IRB) of Palestine Polytechnic University at the following contact: • Email: dsr@ppu.edu (Dr. Mahmoud Alhaddad, IRB Chair).Data will be archived for a minimum of 10 years in accordance with institutional policies. External researchers must submit a formal request outlining the purpose of data use, methodology, and ethical compliance. Approval will be granted by the IRB based on alignment with the original study’s ethical framework.”.

3. Uploaded as supplementary information.

**Additional Editor Comments (if provided):**
**Reviewers' Comments:**

**Comments to the Author**

1. Does this manuscript meet PLOS Digital Health’s publication criteria?

Reviewer #1: Yes

Reviewer #2: Yes

2. Has the statistical analysis been performed appropriately and rigorously?

Reviewer #1: No

Reviewer #2: Yes

3. Have the authors made all data underlying the findings in their manuscript fully available (please refer to the Data Availability Statement at the start of the manuscript PDF file)?

Reviewer #1: No

Reviewer #2: Yes

4. Is the manuscript presented in an intelligible fashion and written in standard English?

Reviewer #1: Yes

Reviewer #2: Yes

Reviewer #1: The topic is relevant, especially given the fragile health-system context in Palestine, but the manuscript needs substantial clarification and correction before it is suitable for publication.

Major comments:

1. There are inconsistencies between the abstract, results text, and Table 4. For example, age is reported as a significant predictor with β = –0.223 in Table 4, but the abstract states “younger nurses (β = 0.223)” and the discussion describes a “moderate positive correlation.”

2. Confidence intervals and p-values look inconsistent (e.g., some CIs do not include 0 but p-values are non-significant, and vice versa). Please re-run the regression, clearly specify whether coefficients are standardized or unstandardized, and correct the table, abstract, and discussion accordingly.

3. The second section of the questionnaire (9 items, 4 domains) is only briefly described. Please provide more detail about exact items (or a clear pointer to the full instrument in Supplementary File 1), response options, scoring approach, handling of reverse-coded items, and how the global “workflow efficiency” score entered the regression. Please clarify whether the tool was developed in English or translated

4. The study uses convenience sampling from governmental hospitals only. The limitations section mentions some biases, but this needs to be more explicit: selection bias, potential over-representation of nurses more comfortable with technology, and limited generalizability to private or rural facilities.

5. The manuscript does not adequately describe the EHR system(s) in use (vendor or local system, core modules, years since implementation, whether nursing workflows are fully digital).

6. PLOS journals typically require deposition of a minimal de-identified dataset in a public repository unless there is a clear, justified ethical barrier. Please revise the Data Availability statement to explicitly align with journal policy.

Reviewer #2: To ensure your manuscript meets the high standards, here are some constructive recommendations.

o Your study is a quantitative cross-sectional design. To adhere to international reporting standards, please ensure you state compliance with the STROBE guidelines (Strengthening the Reporting of Observational Studies in Epidemiology) and include the completed STROBE checklist as a supplementary file.

o I noticed a discrepancy in how the Age variable is reported. The Abstract suggests a positive effect (beta=0.223), but the regression table (Table 4) shows a negative standardized coefficient (beta= -0.223). Since the discussion confirms that younger nurses (lower age) are associated with higher efficiency, the negative sign in Table 4 is statistically correct for the Age variable. Please ensure the Abstract and Discussion consistently reference either the Age variable (with the negative sign) or the 'Younger Nurses' concept (with the positive implication) to avoid confusing the reader.

o You confirmed that the VIF assumption was met (VIF < 3.0). For added rigor, consider briefly reporting the range of VIF values in the results section or in a note beneath Table 4.

o Your study's most distinctive finding is the high rate of interruptions due to technical issues (50.8% of nurses) rooted in infrastructure (power outages, instability). This is a stronger barrier than the 'system complexity' often reported in developed countries. I suggest dedicating a specific paragraph to elaborating on how these real-life infrastructure challenges uniquely constrain EHR benefits in Palestine, significantly differentiating your work.

o Briefly comment on why demographic factors like Gender (p=0.982) and Years of Experience (p=0.250) were not significant predictors. This shows a complete analysis of the tested variables.

**Do you want your identity to be public for this peer review?** For information about this choice, including consent withdrawal, please see our Privacy Policy

Reviewer #1: No

Reviewer #2: No

**Figure resubmission:**

**Reproducibility:** To enhance the reproducibility of your results, we recommend that authors of applicable studies deposit laboratory protocols in protocols.io, where a protocol can be assigned its own identifier (DOI) such that it can be cited independently in the future. Additionally, PLOS ONE offers an option to publish peer-reviewed clinical study protocols. Read more information on sharing protocols at https://plos.org/protocols?utm_medium=editorial-email&utm_source=authorletters&utm_campaign=protocols

---

## [Decision Letter · Decision Letter 1]

20 Jan 2026

Response to Reviewers
Revised Manuscript with Track Changes
Manuscript
**Journal Requirements:**
**Additional Editor Comments (if provided):**
**Reviewers' Comments:**

**Comments to the Author**

Reviewer #1: All comments have been addressed

Reviewer #2: All comments have been addressed

publication criteria?

Reviewer #1: Yes

Reviewer #2: Yes

3. Has the statistical analysis been performed appropriately and rigorously?

Reviewer #1: Yes

Reviewer #2: Yes

4. Have the authors made all data underlying the findings in their manuscript fully available (please refer to the Data Availability Statement at the start of the manuscript PDF file)?

Reviewer #1: Yes

Reviewer #2: Yes

5. Is the manuscript presented in an intelligible fashion and written in standard English?

Reviewer #1: Yes

Reviewer #2: Yes

Reviewer #1: Decision: Minor Revision

The study is valuable and largely sound, but it needs small, concrete fixes to reporting clarity and consistency before acceptance.

Major comments

1. Please explicitly state how each predictor was coded in the regression (e.g., age and experience as continuous years vs. grouped categories; gender coding; Likert predictors treated as continuous). This is essential for interpretation of the unstandardized coefficients (B) and CIs.

2. Please ensure the manuscript uses one consistent framing, either “Age (negative β)” or “Younger nurses (positive implication)”, but not both. Also confirm the Discussion does not describe a “positive correlation” if the standardized beta for age is negative.

3. The 9-item workflow efficiency instrument has good internal consistency, but please add a brief justification for using a single global mean score. If feasible, add a short factor-structure check (even exploratory) or a rationale for not doing so.

4. Because both predictors (e.g., perceived usability/support) and outcomes are self-reported in the same survey, please acknowledge potential common-method variance and that “workflow efficiency” is perceived rather than objectively measured (e.g., no time-motion or EHR log data).

Minor comments:

1. Double-check percentages in Table 1 (some values appear inconsistent with n).

2. Make sure the reverse-coded technical-interruptions item still reports its mean/SD (or clarify why it is omitted).

3. A few grammatical issues remain.

Reviewer #2: No additional Comments. The authors have successfully addressed previous review concerns regarding statistical inconsistencies and methodological detail.

**Do you want your identity to be public for this peer review?** For information about this choice, including consent withdrawal, please see our Privacy Policy

Reviewer #1: No

Reviewer #2: No

**Figure resubmission:**

**Reproducibility:** To enhance the reproducibility of your results, we recommend that authors of applicable studies deposit laboratory protocols in protocols.io, where a protocol can be assigned its own identifier (DOI) such that it can be cited independently in the future. Additionally, PLOS ONE offers an option to publish peer-reviewed clinical study protocols. Read more information on sharing protocols at https://plos.org/protocols?utm_medium=editorial-email&utm_source=authorletters&utm_campaign=protocols

---

## [Decision Letter · Decision Letter 2]

5 Mar 2026

Impact of Electronic Health Records on Nursing Workflow Efficiency and Predictive Factors in Palestinian Hospitals

PDIG-D-25-00985R2

Dear mr farajalla,

We are pleased to inform you that your manuscript 'Impact of Electronic Health Records on Nursing Workflow Efficiency and Predictive Factors in Palestinian Hospitals' has been provisionally accepted for publication in PLOS Digital Health.

Best regards,

Jia-Lang Xu

Academic Editor

PLOS Digital Health

**Additional Editor Comments (if provided):**

**Reviewer Comments (if any, and for reference):**

Reviewer's Responses to Questions

**Comments to the Author**

Reviewer #1: All comments have been addressed

publication criteria?

Reviewer #1: Yes

3. Has the statistical analysis been performed appropriately and rigorously?

Reviewer #1: Yes

4. Have the authors made all data underlying the findings in their manuscript fully available (please refer to the Data Availability Statement at the start of the manuscript PDF file)?

Reviewer #1: Yes

5. Is the manuscript presented in an intelligible fashion and written in standard English?

Reviewer #1: Yes

Reviewer #1: All comments have been well addressed.

**Do you want your identity to be public for this peer review?** For information about this choice, including consent withdrawal, please see our Privacy Policy

Reviewer #1: No
